# CHRAC/ACF contribute to the repressive ground state of chromatin

Alessandro Scacchetti[1,2] , Laura Brueckner[3,*] , Dhawal Jain[1,2,*], Tamas Schauer[1,2], Xu Zhang[4,5] , Frank Schnorrer[4] , Bas van Steensel[3], Tobias Straub[6] , Peter B Becker[1,2]

The chromatin remodeling complexes chromatin accessibility complex and ATP-utilizing chromatin assembly and remodeling factor (ACF) combine the ATPase ISWI with the signature subunit ACF1. These enzymes catalyze well-studied nucleosome sliding reactions in vitro, but how their actions affect physiological gene expression remains unclear. Here, we explored the influence of *Drosophila melanogaster* chromatin accessibility complex/ACF on transcription by using complementary gain- and loss-of-function approaches. Targeting ACF1 to multiple reporter genes inserted at many different genomic locations revealed a context-dependent inactivation of poorly transcribed reporters in repressive chromatin. Accordingly, single-embryo transcriptome analysis of an *Acf* knock-out allele showed that only lowly expressed genes are derepressed in the absence of ACF1. Finally, the nucleosome arrays in *Acf*-deficient chromatin show loss of physiological regularity, particularly in transcriptionally inactive domains. Taken together, our results highlight that ACF1-containing remodeling factors contribute to the establishment of an inactive ground state of the genome through chromatin organization.

## Introduction

The chromatin accessibility complex (CHRAC) and the related ATP-utilizing chromatin assembly and remodeling factor (ACF) are prototypic nucleosome sliding factors purified originally from extracts of *Drosophila melanogaster* embryos ([1], [2]). ACF consists of ISWI, an ATPase of the helicase superfamily 2, and a large subunit, ACF1. ACF associates with two histone-fold subunits, CHRAC-14 and CHRAC-16, to form CHRAC ([3]). Both complexes have very similar nucleosome sliding activity in vitro ([4]). Because ISWI is present in several other nucleosome remodelers ([5]), ACF1 serves as the signature regulatory subunit for the two complexes.

The mechanism of nucleosome sliding has been well described by biochemical and biophysical studies. ISWI and ACF1 bind target nucleosomes and flanking linker DNA. Substrate binding and ATP hydrolysis cycles trigger conformation changes in the remodeler that disrupt histone–DNA interactions and eventually displace the intact histone octamer along the DNA, effectively sliding a nucleosome ([6], [7], [8], [9], [10], [11], [12]) (for review, see reference [13]).

Nucleosome sliding may theoretically affect transcription through local and global mechanisms ([14]). Locally, nucleosomes could be slid off promoters, exposing binding sites for transcription factors. Conversely, a remodeler might push nucleosomes to occlude regulatory sequences. The yeast Isw2 complex, which is related to the metazoan CHRAC complexes, has been shown to slide nucleosomes toward promoters ([15], [16]). Alternatively, nucleosome sliding factors may influence transcription by globally affecting the tightness of DNA packaging in chromatin. Nucleosome sliding factors may improve the regularity of nucleosome arrays by closing gaps ("nucleosome spacing"), thus minimizing the level of accessible DNA ([1], [6], [17]). In vitro, regularly spaced nucleosome arrays readily fold into "30 nm"–type fibers, a process that has been suggested to promote the formation of "higher order," repressive chromatin structures ([14]).

A role for CHRAC/ACF in establishing such repressive chromatin had been derived from early studies of *Acf* mutant embryos that documented defects in nucleosome spacing, in the formation of repressive pericentric heterochromatin and polycomb-mediated silencing ([18], [19]). A more direct role for ACF1 in the repression of wingless target genes has also been described ([20]). The phenotypic oogenesis defects observed in *Acf* mutants ([21]) may be explained by either mechanism. These early

[1]Molecular Biology Division, Biomedical Center, Faculty of Medicine, Ludwig-Maximilian University Munich, Planegg-Martinsried, Germany   [2]Center for Integrated Protein Science Munich, München, Germany   [3]Division of Gene Regulation, Netherlands Cancer Institute, Amsterdam, The Netherlands   [4]Developmental Biology Institute of Marseille, Aix Marseille University, Centre Nationnal de la Recherche Scientifique, Marseille, France   [5]School of Life Science and Engineering, Foshan University, Foshan, China   [6]Bioinformatic Unit, Biomedical Center, Faculty of Medicine, Ludwig-Maximilian University Munich, Planegg-Martinsried, Germany

Correspondence: pbecker@bmc.med.lmu.de
*Laura Brueckner and Dhawal Jain contributed equally to this work.

studies based their conclusions on the analysis of *Acf¹* and *Acf²* alleles, which were later shown in the context of oogenesis not to deliver a complete loss-of-function genotype because they still express the C-terminal PHD/bromo domains of ACF1 (21). Indeed, some oogenesis phenotypes observed in *Acf¹* and *Acf²* could not be reproduced with a larger gene deletion (*Acf⁷*, considered a true null allele) or under RNAi conditions, as shown in reference 21. Therefore, the consequences of a complete loss of ACF1 (and thus the remodeling complexes it defines) are still unknown.

Clues about ACF1 functions at specific loci may be derived from mapping the chromosomal binding sites of the remodeler by chromatin immunoprecipitation (ChIP). Unfortunately, despite many efforts, we were not able to map ACF1 binding sites by ChIP, presumably because the interaction of the remodeler is too transient and dynamic to be efficiently cross-linked (22).

To unequivocally clarify the effect of ACF1-containing remodelers on transcription, we performed two key experiments. In a gain-of-function approach, we artificially targeted ACF1 to reporter genes integrated at many different chromatin loci and monitored the consequences for reporter gene transcription. Furthermore, using a null allele, we compared the transcriptome of individually staged null mutants to that of matched wild-type embryos. Both approaches suggest that the main effect of ACF1 on transcription is that it participates in the silencing of genes in inactive chromosomal domains. Importantly, derepression in mutant embryos correlates with defects in nucleosome spacing. Hence, we conclude that ACF1-containing remodelers contribute to a repressed ground state of the genome through chromatin organization.

# Results

## Artificial ACF1 tethering leads to context-dependent repression

To investigate potential effects of ACF1 on transcription, we first applied an established approach involving the ectopic targeting of ACF1 to a reporter gene locus. We used a fly line with a defined genomic insertion of a reporter gene cassette consisting of *lacZ/mini-white* genes and 5' UAS^Gal sequences (23) (Fig 1A). We generated flies expressing N- or C-terminal fusions of ACF1 to the DNA-binding domain of the yeast activator GAL4 (GAL4DBD) under the control of the endogenous *Acf* promoter, which assured expression at levels comparable to endogenous ACF1 (Fig 1A and B). A control line harbored a construct expressing only the GAL4DBD. Mating the two types of fly lines yields offspring in which ACF1 is recruited to the UAS^Gal element.

Successful tethering of ACF1-GAL4DBD in early embryos was confirmed by ChIP–quantitative PCR (qPCR) (Fig 1C). The entire remodeler seems to be recruited because its catalytic partner ISWI was also detected at the UAS sites (Fig S1A). ACF1 targeting resulted in an about twofold reduced *LacZ* transcription relative to the GAL4DBD only (mean fold-change = 0.58 for GAL4DBD-ACF1 and mean fold-change = 0.39 for ACF1-GAL4DBD) (Fig S1B). However, no obvious changes in nucleosome positions over and around the reporter locus could be scored by MNase-seq (Fig S1C and D).

These experiments provided the proof of principle that a functional ACF1-GAL4DBD fusion protein could be recruited to UAS^Gal elements integrated in the fly genome, but these lacked the necessary generalization to document the presumed repressive effect. Therefore, we used a previously characterized library of several

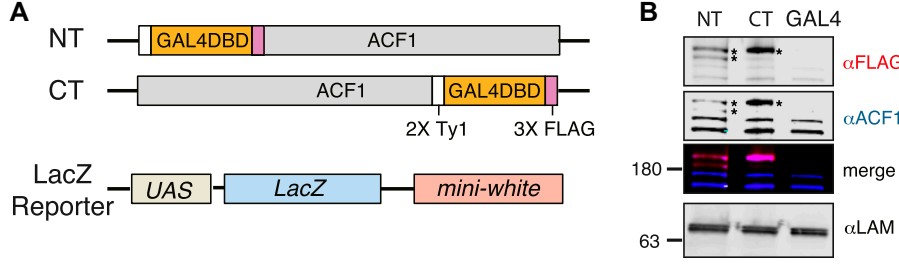

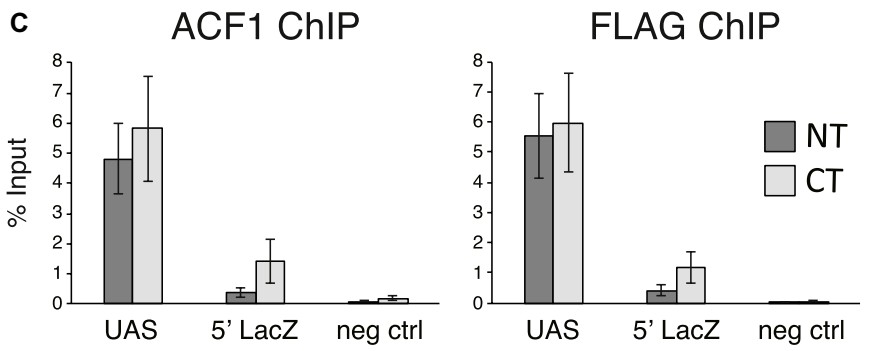

**Figure 1.  ACF can be tethered to a reporter locus through GAL4-DBD.**
**(A)** Schematic illustration of the transgenes used for testing the effects of ACF1 recruitment. ACF1 is fused to the GAL4 DNA-binding domain (GAL4DBD) at either the N-terminus or the C-terminus. A transgene containing the GAL4DBD alone (GAL4) is used as a negative control. The reporter transgene contains five UAS^Gal4 (UAS) 5' of *LacZ* and *mini-white* genes. **(B)** Western blot detection of ACF1 in an embryo nuclear extract (0–16 h AEL). Endogenous and fusion proteins were detected with a specific ACF1 antibody (blue channel); the ACF1-GAL4 fusions are FLAG-tagged and detected with an anti-FLAG antibody (magenta channel). Asterisks indicate the expressed transgenic ACF1-GAL4 fusions. Embryos containing a transgene coding for the GAL4DBD alone (GAL4) are included as a negative control. Lamin serves as a loading control. **(C)** ChIP-qPCR monitors the recruitment of ACF1 to UAS in 0- to 12-h embryos. The immunoprecipitation was conducted using ACF1 and FLAG antibodies. "UAS" and "5' LacZ" denote the regions amplified by qPCR. Bars denote average % Input enrichment (n = 3 biological replicates) ± SEM. "neg ctrl" represents a negative control locus (encompassing the *Spt4* gene).

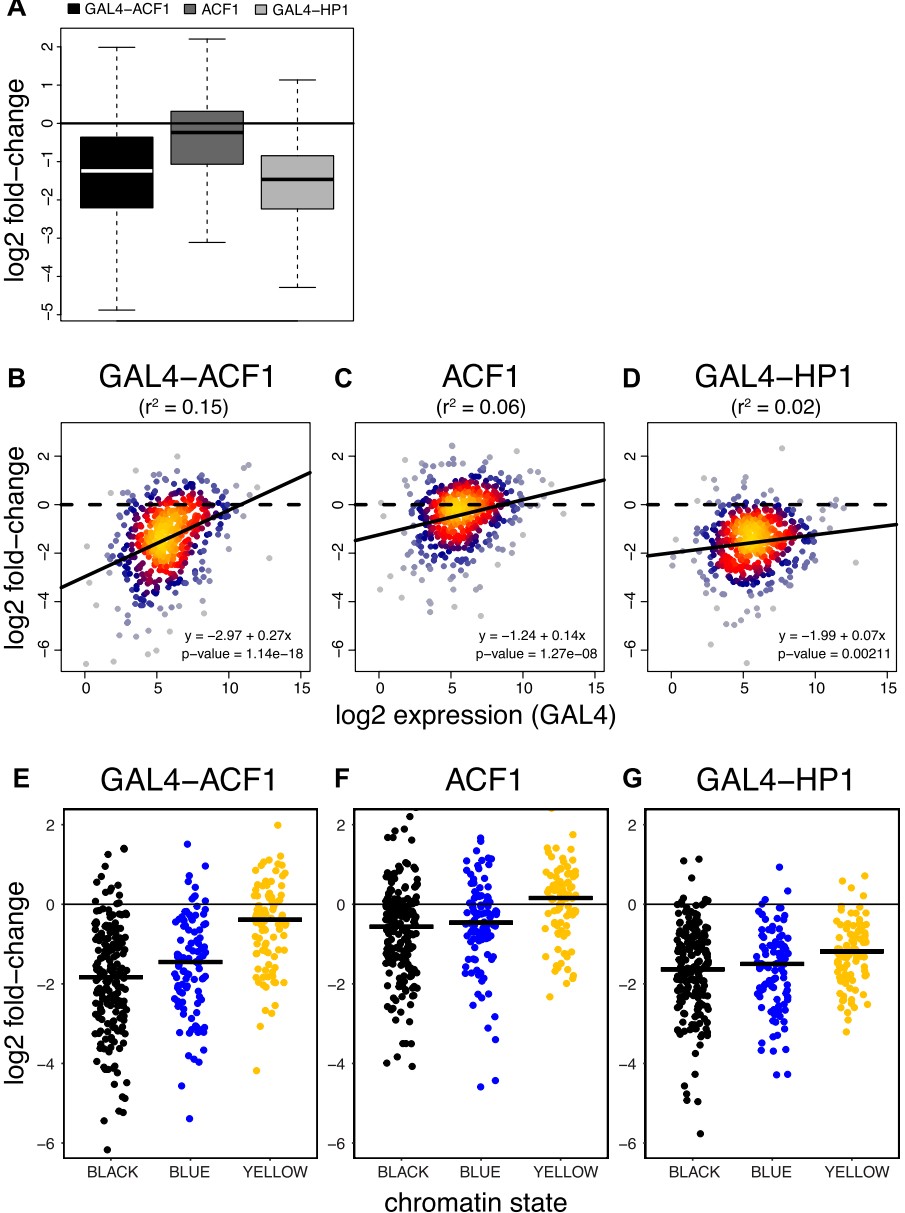

**Figure 2. ACF1 represses multiple reporters in a context-dependent manner.**
**(A)** Boxplots represent log2 fold-change distribution upon ACF1 tethering (GAL4-ACF1), ACF1 overexpression (ACF1), or HP1 tethering (GAL4-HP1), compared with the control (N = 492). **(B)** Log2 fold-change for each reporter in relation to its mean log2 expression upon ACF1 tethering versus the control (GAL4-ACF1) (N = 492). Black lines represent linear regression fit. $r^2$ values derived from the linear model are shown in parentheses. Equations of the regression lines are displayed in the plot. *P*-value refers to the significance of this relationship (slope). **(C)** Same as (B) but for ACF1 overexpression (ACF1). **(D)** Same as (B) but for HP1 tethering (GAL4-HP1). **(E)** Jitter plots represent the distribution of log2 fold-changes of reporters integrated in BLACK (N = 197), BLUE (N = 102), and YELLOW (N = 94) chromatin domains for the case of tethered ACF1 (GAL4-ACF1). Black horizontal bars represent median. **(F, G)** Same as (E) but for ACF1 overexpression (ACF1) and HP1 tethering (GAL4-HP1), respectively.

hundred barcoded reporter genes that had been randomly integrated into the genome of *Drosophila* Kc167 cells. We previously tethered heterochromatin protein 1 (HP1) as a GAL4DBD fusion to these sites and determined how the chromatin environment modulated HP1 repression (24). HP1, a known repressor, provides a convenient reference for ACF1 in this system. In parallel transient transfections, we introduced the various constructs into Kc167 cells and confirmed their expression by Western blotting and immuno-fluorescence microscopy: GAL4-ACF1, tagged ACF1 lacking a GAL4DBD, ACF1 lacking any tag, and a tagged GAL4DBD (Fig S2A–C).

As in the case of the single-reporter system, recruitment of ACF1 resulted in a general down-regulation (median log2 fold-change = −1.24), almost comparable to HP1 (median log2 fold-change = −1.46) (Fig 2A), which served as a positive control. Expression of ACF1 lacking

the GAL4DBD had a much weaker effect than its tethered counterpart (median log2 fold-change = −0.24). Interestingly, the extent of ACF1-induced repression inversely correlated with the mean expression levels of the reporters: the repressive effect was less pronounced for reporters with a high expression level (Fig 2B). A similar correlation could be also observed for the untethered ACF1 (Fig 2C). The tethered HP1 showed instead just a small correlation between down-regulation and reporter expression (Fig 2D), significantly different from GAL4-ACF1 (Fig S2D).

To explore whether the chromatin environment in which the individual reporter genes are integrated modulates ACF1-mediated repression, we referred to the five-state model of chromatin (25). In this model, YELLOW and RED represent constitutively and developmentally regulated active chromatin domains, respectively;

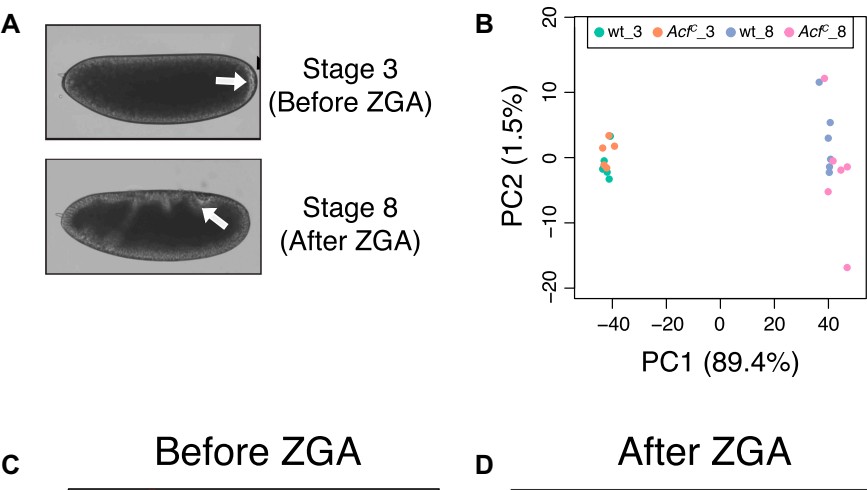

**Figure 3. ACF1 loss perturbs gene expression in early embryos.**
**(A)** Embryo stages selected for transcriptome analysis. In Bownes Stage 3, zygotic transcription is not established yet. Bownes Stage 8 shows robust zygotic transcription. Arrows highlight morphological features of the corresponding stages (appearance of pole cells in Stage 3 and germ band elongation in Stage 8). **(B)** PCA of single-embryo transcriptomes. Each dot represents a single replicate for the corresponding genotype/condition. _3 and _8 indicate embryos before and after ZGA, respectively. **(C)** Differential gene expression analysis of coding genes from RNA-seq data. Wild-type and $Acf^C$ transcriptomes were compared before ZGA (N = 7,585). Scatter plots represent log2 fold-change of $Acf^C$ over wild type for each gene in relation to its mean expression (mean of normalized counts). Red dots represent significant (q-value < 0.1) up- or down-regulated genes. **(D)**. Same as (C) but after ZGA (N = 10,088).

GREEN corresponds to HP1-marked heterochromatic domains; and BLACK and BLUE correspond to inactive and polycomb-repressed domains, respectively. We added a sixth state, GRAY, to refer to reporters integrated in genomic regions not defined by any of the original five states. We found that reporters integrated in BLACK and BLUE chromatin domains are strongly repressed upon ACF1 targeting (BLACK: median log2 fold-change = −1.83; BLUE: median log2 fold-change = −1.45), whereas the ones integrated in YELLOW are only slightly affected (median log2 fold-change = −0.39) (Fig 2E). Similarly, down-regulation of reporters in RED (median log2 fold-change = −0.70) and GREEN (median log2 fold-change = −0.97) states results smaller than the one in BLACK and BLUE domains (Fig S2E). Recruitment of HP1, instead, shows a more general repressive effect compared with that of ACF1, which does not correlate with any type of chromatin (Figs 2G and S2G). Interestingly, expression of untethered ACF1 also shows a very mild context-dependent repression, reminiscent of its tethered counterpart (Figs 2F and S2F).

In summary, the tethering approach suggested that ACF1-containing remodelers have a repressive function. In contrast to repression by HP1, which served as a positive control for repression, the effect of ACF1 strongly depends on the chromatin context and is particularly robust in lowly expressed genes in overall inactive chromatin domains.

## Transcriptome analysis of *Acf*-deficient *Drosophila* embryos

The tethering experiment suggested that ACF1 may not affect gene transcription like a classical corepressor, but at a more fundamental level. However, given the artificial nature of the approach with its uncertainties about the functionality of the DBD fusion protein and the remodeling activity of the reconstituted complex, we sought to test the hypothesis of a context-dependent repressive effect of ACF1 by using a loss-of-function approach in a physiological system. A transcriptome analysis for an ACF1 deficiency has not been reported so far. Conceivably, a function of ACF/CHRAC may be best observed during early embryogenesis in *Drosophila* because ACF1 expression peaks during these stages and both CHRAC and ACF have been originally identified in embryos. Early on, embryogenesis defects had been noted for the $Acf^1$ allele (18). However, this allele only deletes an N-terminal fragment of the *Acf* gene and still allows the expression of a C-terminal "stub" containing a PHD/bromo domain module that may interfere with relevant interactions. We later concluded that the more extensive deletion of the $Acf^7$ allele most likely represents a clean loss of function (21). In parallel with these earlier studies, we generated a clean *Acf* gene deletion using a CRISPR/Cas9-based engineering approach ($Acf^C$). Expression of ACF1 is not detectable by Western blotting in homozygous embryos for the $Acf^C$ or $Acf^7$ alleles (Fig

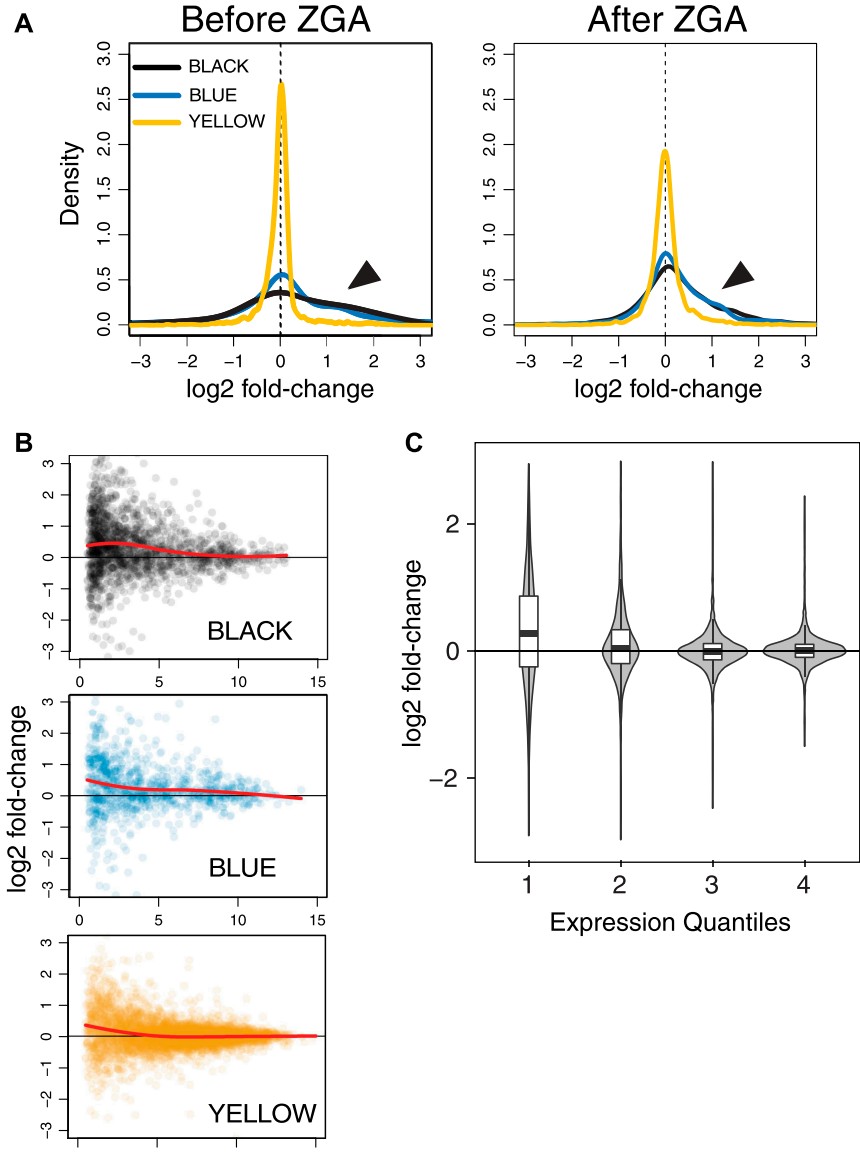

**Figure 4. Loss of ACF1 affects transcription prominently in inactive chromatin.**
**(A)** Comparison of wild-type and *Acf^C* transcriptomes in relation to the five-state chromatin model. Plots represent the distribution of log2 fold-changes for genes belonging to the YELLOW, BLUE, and BLACK chromatin domains before and after ZGA. Arrows indicate the differences between BLACK/BLUE and YELLOW. **(B)** Each scatter plot represents log2 fold-change for each gene of the indicated chromatin state in relation to its mean expression (after ZGA only). Colors match the chromatin domains as described in the five-state model. Red lines represent local regression fit. **(C)** Violin plots represent log2 fold-change distributions for each given expression quartile (after ZGA only), regardless of the chromatin state. Boxplots are overlapped to show median values.

S3A) (21). *Acf^C* and *Acf^7* embryos show a slightly lower hatching rate compared with their wild-type counterparts (Fig S3B; unpublished observation), but the survivors develop normally into viable and fertile flies (unpublished observation). We were concerned that *Acf^C* mutants might develop slower than wild-type embryos and hence did not rely on simple developmental stage timing for proper transcriptome comparison. Rather, we selected single embryos either before zygotic genome activation (ZGA) or after ZGA based on morphological hallmarks (Fig 3A; see the Materials and Methods section) and determined their transcriptome by RNA-seq analysis. Principal component analysis (PCA) showed no strong differences between *Acf^C* and wild type in both developmental stages (Fig 3B) with clear transition from the maternal to the zygotic RNA pool (Fig S3C and Table S1). Differential gene expression analysis revealed a relatively small number of genes significantly affected by ACF1

loss at both stages (Fig 3C and D, and Table S2), but without a clear direction (activation or repression) and without a uniquely defined gene ontology enrichment (Fig S3D).

## Deletion of the *Acf* gene leads to relaxation of the repressive ground state of chromatin in early embryos

The relatively small number of differentially expressed genes upon *Acf* deletion may be explained by functional redundancy with other remodelers. However, the observation of context-dependent effects of ACF1 tethering prompted us to relate the transcription effects in embryos to the chromatin state of genes and to their transcriptional activity.

Evaluating the differences between *Acf^C* and wild-type embryos in the context of the five-state model of chromatin organization, we

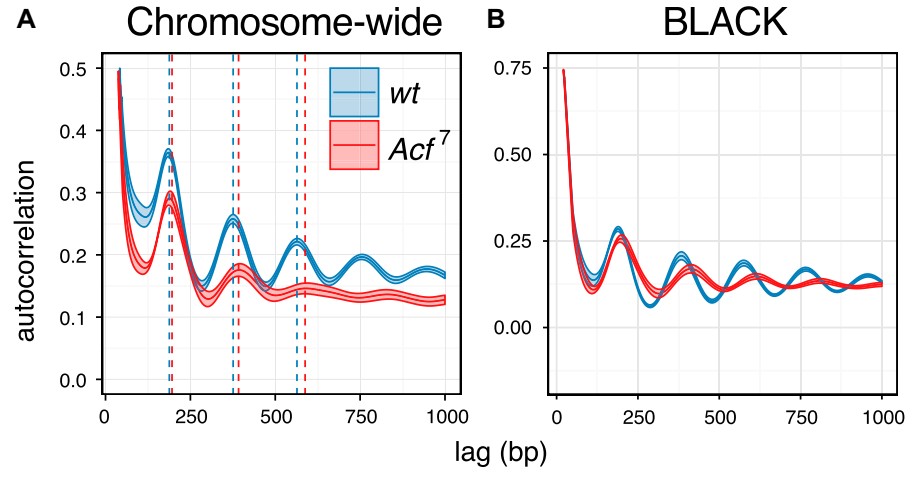

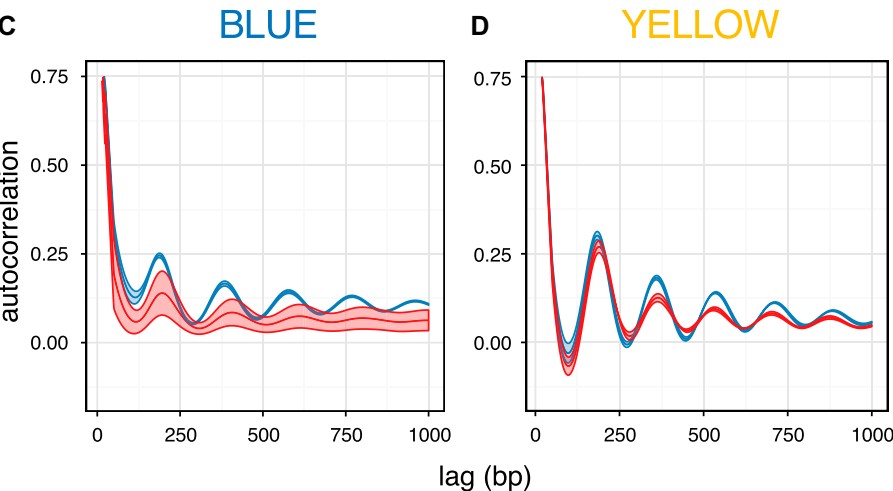

**Figure 5.   Global and context-dependent decrease in nucleosome regularity as a consequence of ACF1 loss.** **(A)** Changes in nucleosome periodicity on chromosomes 2 and 3 are estimated using the autocorrelation function. The correlation coefficients for the nucleosome occupancy values are plotted against the relative shifts (lag). The mean and SEM of replicate samples (n = 5 for wt and n = 3 for $Acf^7$) are displayed. Dashed lines indicate the centers of nucleosome positions derived from the autocorrelation peaks. **(B)** Changes in nucleosome periodicity in BLACK chromatin domains estimated by the autocorrelation function. The mean and SEM of replicate samples are displayed. **(C, D)** Same as (B) but for BLUE and YELLOW chromatin domains, respectively.

observed a small increase in expression of genes in inactive BLACK and BLUE chromatin domains in *Acf*-deficient embryos. In contrast, genes in the YELLOW domains (Fig 4A) and in GREEN or RED chromatin (Fig S4A) were largely unaffected by the loss of ACF1. We correlated our data with the modENCODE histone modification data (26) that had been obtained from 2- to 4-h embryos, very close to the zygotic stage analyzed in our study. The derepression of transcription upon ACF1 loss correlates with the absence of defined chromatin marks (BLACK), with the presence of H3K27me3 (BLUE), and with the absence of H3K36me3 (YELLOW) (Fig S4B, top panels). No clear correlation was observed for H3K9me3 (GREEN) or H3K4me3 (RED) (Fig S4B, bottom panels). The ACF1-dependent effect was most pronounced for lowly expressed genes, not only in the BLACK and BLUE domains but also in active YELLOW chromatin (Fig 4B). Indeed, the extent of derepression in *Acf* mutant embryos correlates generally with low expression levels, regardless of the chromatin domain a gene resides in (Fig 4C).

We conclude that loss of ACF1 leads to a widespread derepression of genes that are characterized by low levels of transcription in wild type. This supports our earlier conclusion derived from the ACF1 tethering experiments.

**CHRAC/ACF repress inactive chromatin by maintaining nucleosome regularity**

Nucleosome sliding by CHRAC/ACF improves the regularity of nucleosome arrays in vitro and hence optimizes the packaging of DNA. Lack of nucleosome spacing activity in vivo leads to irregular chromatin, which may explain the observed derepression phenotype in *Acf* mutants. To test this hypothesis, we analyzed high-quality nucleosome occupancy maps obtained from wild-type and $Acf^7$ embryos (27 Preprint) and assessed global chromatin regularity by applying an autocorrelation function to the nucleosome dyad density patterns. Briefly, the function calculates the correlation between nucleosome dyad signals in an array of nucleosomes with a stepwise-shifted copy of itself. The calculated correlation coefficients for each shift (lag) are then plotted as a function of the shift (lag) length. Autocorrelation has previously been applied to score nucleosome repeat lengths (28) and promoter architecture (29). Applied to nucleosome maps, the analysis reveals periodic oscillations, in which the amplitude and the decay rate provide information about regularity, whereas the maxima reveal the distance between adjacent nucleosomes. For $Acf^7$ embryos, the analysis documents a genome-wide decay of autocorrelation amplitude together with

a trend toward increased nucleosome repeat length (wild type = 188.4 ± 0.7 bp, $Acf^7$ = 195.3 ± 1.5 bp) (Fig 5A). Evidently, loss of ACF1 globally affects the regularity and spacing of nucleosome arrays. To determine whether this global trend applied to the five chromatin states, the autocorrelation analysis was repeated for each of the chromatin domains. First, we found a dampening of the autocorrelation function upon ACF1 loss in the BLACK and BLUE inactive domains, which was not evident for the YELLOW active domains (Fig 5B–D). Second, we found a context-dependent increase in nucleosome repeat length in BLACK (wild type = 192.6 ± 0.5 bp, $Acf^7$ = 206.0 ± 0.6 bp) and BLUE (wild type = 192.2 ± 0.6 bp, $Acf^7$ = 201.0 ± 2.5 bp) but not in YELLOW (wild type = 182.0 ± 0.5 bp, $Acf^7$ = 184.0 ± 1.2 bp) domains.

The correlation between the decay of physiological chromatin regularity and derepression of transcription in *Acf* mutant embryos suggests that the reduced stringency of DNA packaging in the absence of prominent spacing factors perturbs the repressed ground state of the genome installed by nucleosome arrays.

## Discussion

CHRAC was identified two decades ago following a biochemical activity that increased the accessibility of DNA in in vitro assembled chromatin (1). Further characterization revealed that CHRAC did not destroy chromatin, but, to the contrary, improved the regularity of nucleosome fibers, identifying a first nucleosome spacing factor. This conundrum was resolved by the discovery that ISWI-containing remodeling factors catalyze nucleosome sliding (6). ACF, which slides nucleosomes such as CHRAC, was originally purified searching for chromatin assembly factors (17, 30). The first genetic analyses of *Acf* deficiencies highlighted defects in pericentric heterochromatin and suppression of variegation, supporting the idea that both "higher order" chromatin structures and gene silencing rely on proper chromatin organization (18, 19). To date, however, a systematic assessment of the contribution of ACF1-containing remodelers to transcription has not been performed.

Our current study now clarifies this open issue. In our experimental design, we avoided several potential pitfalls. (i) We used clear *Acf* gene deletion with a clean null phenotype. Previous studies used $Acf^1$ and $Acf^2$ alleles that were later shown to yield oogenesis phenotypes that were clearly distinct from true loss-of-function phenotypes (21). (ii) To assure that the transcriptome analysis was not flawed by a delay in the development of mutant embryos, we hand-selected mutant and wild-type embryos of matched age and determined their transcriptomes individually. (iii) We used two orthogonal approaches, each avoiding the technical or conceptual shortcomings of the other.

Although we could confirm a function of CHRAC/ACF in gene silencing, the extent of transcriptional repression scored in our tethering system was much stronger compared to the one in developing embryos. However, the consequences of the genetic deficiency may be masked by functional redundancy. For example, the ISWI-containing RSF remodeling complex (31, 32) possesses similar nucleosome assembly and spacing activities as CHRAC/ACF. The targeting of ACF1 via an ectopic DNA-binding domain is expected to locally increase the ACF1 concentration around the tethering site, allowing effects to be scored above the background activities of endogenous factors. Regardless of magnitude, both types of experiments yielded highly complementary results.

The high-throughput targeting system we employed has previously been validated for HP1 (24), a well-known repressor, which provided an important benchmark. The repression induced by ACF1 recruitment was of the same order of magnitude as the effect of HP1 tethering determined in parallel. However, the repression mediated by targeted ACF1 was strongly modulated by the chromatin environment, with an obvious effect in overall inactive chromatin domains and lowly expressed genes. Nucleosome remodelers can work if tethered (33), but given the dynamic nature of DNA interactions observed with most transcription factors, we think that the tethering rather increased the local concentration of the factor around the $UAS^{Gal}$ site. In support of this notion, the overexpression of ACF1 lacking a DNA-binding domain had a similar effect, but milder. Importantly, the selective effect of ACF1 on poorly transcribed genes was similar, whether the ACF1 concentrations were increased globally or locally. This context dependence of CHRAC/ACF repression was confirmed by studying *Acf* deficiency in developing embryos.

Notwithstanding possible functional redundancies, we detected a major and global impact of ACF1 on physiological nucleosome regularity by applying an autocorrelation function to genome-wide nucleosome dyad maps. The impact of ACF1 depletion was more evident for inactive chromatin domains, establishing a clear correlation between the extent of physiological chromatin regularity and general repression, which we suggest is of a causal nature.

Various ISWI-type nucleosome sliding factors have very different functions. NURF (30, 34), for example, is recruited by sequence-specific transcription factors to promoters of certain gene classes, where it serves as a coactivator (35). CHRAC/ACF, in contrast, are most likely not targeted to promoters and enhancers (22). Conceivably, these remodelers might establish the regularity of the nucleosome fiber in the context of replication (19, 36) and/or DNA repair (37, 38) and may exert a general "surveillance" function in search for gaps in the nucleosome array to be closed. We propose that their action establishes a repressive ground state of chromatin, rendering the genome inaccessible through optimal nucleosome packaging. Any further regulation, such as the specific activation of genes by recruitment of histone modifiers and more dedicated remodelers as well as the targeting of silencing machineries, happens on top of the general naive infrastructure provided by regular nucleosome arrays. In support of this idea, *Acf* depletion did not significantly affect the expression of transposable elements (unpublished observation), which are silenced through heterochromatinization. CHRAC/ACF and related factors are to be considered the caretakers of this genomic infrastructure. Their important and global role in generating a basal level of genome-wide repression can only be appreciated in regions that are devoid of all other, more potent, targeted and specific regulatory mechanisms.

## Materials and Methods

### *Drosophila* strains and genetics

The ACF1-GAL4 fusion constructs were generated by recombineering (39). Briefly, a fosmid containing the genomic region of *Acf*

(pflyfos021945) was recombined in *Escherichia coli* with a combinatorial tag cassette consisting of 2x-TY1-GAL4DBD(1-147)-3XFLAG to tag ACF1 either at its N-terminus or its C-terminus, or to entirely replace its coding sequence to serve as a control. Fosmids were inserted into attp40 (*yw*; attP40, locus 25C7, chr2L) (Genetic Services Inc.) to generate fly lines with ACF1 transgenes in chromosome 2L. The mosaic F0 generation was crossed with *w1118* and progeny flies from generation F1 onward were screened for dsRed phenotype (red eye fluorescence). Homozygous stocks were established by tracking eye fluorescence and the expression of ACF1 constructs was confirmed by Western blotting. ACF1-GAL4 transgenic flies were crossed to N1 flies (containing the *UAS-LacZ-mini-white* reporter (23)), to generate the final tethering system. The *Acf⁷* allele had been described earlier (21). It contains a deletion of most of the *Acf1* coding sequences and, to the best of our knowledge, corresponds to a loss-of-function phenotype.

### Generation of the *Acf^C* mutant allele

Predicted single guide RNA (sgRNA) targeting sequences for 5′ and 3′ ends of ACF1 were obtained from the Zhang lab CRISPR resource (a total eight for 5′ end and four for 3′ end) (http://crispr.mit.edu/). The 20-bp targeting sequences were inserted into the framework of primer-1 (5′-TAATACGACTCACTATAG-(targeting sequence)-GTTTTA-GAGCTAGAAATAGC-3′) in the 5′ to 3′ direction. Using a scaffold primer (5′-AAAAGCACCGACTCGGTGCCACTTTTTCAAGTTGATAACGGAC-TAGCCTTATTTTAACTTGCTATTTCTAGCTCTAAAAC-3′) and a universal reverse primer (5′-AAAAGCACCGACTCGGTGCC-3′), a final DNA was assembled for in vitro transcription by PCR. The PCR product was purified using the GeneElute PCR cleanup kit (Sigma, Cat. No. NA1020). In vitro transcription was performed using the T7 MEGA-shortscript kit (Ambion, Cat. No. AM1354) and purified RNA was assessed by agarose gel electrophoresis.

Efficiency of the RNA-mediated cleavage was assessed by transfecting 1 μg sgRNA to $7 \times 10^5$/ml SL2 cells (clone Hgr14 stably expressing Cas9) (40) in 2 ml final volume (24-well plate). Genomic DNA was prepared after 48 h. An ~600-bp region surrounding the selected guide RNA (gRNA) sequences was amplified, and the PCR product was melted at 95°C for 5 min and then cooled slowly at the ramp rate of 0.1°C/s (41). gRNA cleavage frequently gives rise to mismatched base pairs around the cutting site, which were detected by T7 endonuclease (M0302S, NEB) cleavage and agarose gel electrophoresis. gRNA combinations that lead to T7 endonuclease cleavage were selected.

Genomic DNA 1.3 kb upstream and 1.5 kb downstream of gRNA sequences for ACF1 were amplified using a high-fidelity PCR system. These homology arms excluded the sgRNA sites. The homology arms and 3XP3-dsRed fly selection cassette (obtained from pJet1.2 (41)) were assembled together in a pBS donor backbone by using the Golden Gate cloning strategy. The final clone was validated by sequencing.

The purified plasmid and sgRNA for 5′ and 3′ ends of the *Acf* gene were co-injected into blastoderm embryos of *yw; Cas9; lig⁴¹⁶⁹* genotype (42). The F0 mosaic males were crossed with *w1118* females and F1 transformants were screened for red fluorescence eye phenotype. The flies were backcrossed to the *yw* strain for four subsequent generations and rendered homozygous. Deletion of the

locus was screened by PCR and loss of protein was assessed on Western blot. Final deletion of ACF1 encompasses around 4 Kb from the 562$^{nd}$ base onward, removing most of the gene except its 5′ and 3′ UTRs. No ACF1 protein could be detected in the newly generated *Acf^C* mutant (see the Results section), similarly to that observed in the previously analyzed *Acf⁷* allele (generated by imprecise p-element excision) (21).

For hatching assays, 0- to 16-h embryos were collected on apple juice agar plates and allowed to develop for an additional 25 h at 25°C. Hatched larvae were counted.

### Nuclei isolation and Western blot

For isolation of nuclei, embryos were collected overnight (0–16 h after egg laying [AEL]) onto apple juice agar plates and dechorionated in 25% bleach for 5 min. After extensive washes with PBS (140 mM NaCl, 2.7 mM KCl, 10 mM $Na_2HPO_4$, and 1.8 mM $KH_2PO_4$), the embryos were transferred to 1.5-ml tubes, resuspended in NB-0.3 (15 mM Tris–Cl, pH 7.5, 60 mM KCl, 15 mM NaCl, 5 mM $MgCl_2$, 0.1 mM EGTA, pH 8, 0.3 M sucrose, 0.2 mM PMSF, 1 mM DTT, and Roche cOmplete protease inhibitor without EDTA), and homogenized using a metal pestle (LLG Labware, Cat. No. 9.314.501). The homogenate was collected and carefully layered on top of a biphasic solution consisting of NB-1.4 (15 mM Tris–Cl, pH 7.5, 60 mM KCl, 15 mM NaCl, 5 mM $MgCl_2$, 0.1 mM EGTA, pH 8, and 1.4 M sucrose) and NB-0.8 M sucrose. After spinning at 13 krpm for 10 min (4°C), the nuclei pellet was collected and washed twice with NB-0.3 (spinning at 5,000 rpm for 5 min at 4°C in between washes).

For Western blot analysis, the nuclei were suspended in 5× Laemmli sample buffer (250 mM Tris–HCl, pH 6.8, 10% w/v SDS, 50% v/v glycerol, 0.1% w/v bromophenol blue, and 10% β-mercaptoethanol) and boiled at 96°C for 8 min. The following antibodies were used for Western blots: αACF1 8E3 (22) (1:5), αFLAGm2 (1:1,000, Sigma, Cat. No. F1804), and α Lamin T40 (1:1,000, a kind gift from H. Saumweber).

### ChIP-qPCR

For ChIP analysis, embryos were collected 0–12 h AEL and dechorionated in 25% bleach for 3 min. After extensive washes with water, the embryos were transferred to 15-ml tubes and weighted. Between 0.5 and 1 g of embryos were washed with 50 ml of PBS/ 0.01% Triton X-100 and then resuspended in 9 ml of fixing solution (50 mM Hepes, pH 7.6, 100 mM NaCl, 1 mM EDTA, and 0.5 mM EGTA)/ 3.7% formaldehyde (Merck, Cat. No. 1040031000). Then 30 ml of n-heptane was added and the tubes were shaken for 1 min, followed by 13.5 min of incubation on a rotating wheel (18°C). The embryos were pelleted at 3,000 rpm for 1 min, resuspended in 50 ml of PBS/0.01% Triton X-100/125 mM glycine, and incubated at RT for 5 min. After two washes with PBS/0.01% Triton X-100, the embryos were frozen in liquid nitrogen and stored at −80°C until further processing. The frozen embryos were resuspended in 5 ml of RIPA buffer (10 mM Tris–Cl, pH 8, 1 mM EDTA, 140 mM NaCl, 1% Triton X-100, 0.1% SDS, and 0.1% sodium deoxycholate/1 mM DTT/0.2 mM PMSF/ Roche cOmplete Protease inhibitor without EDTA) and dounced 10 times using a loose pestle and 10 times with a tight pestle. The homogenate was transferred to a 15-ml tube and spun at 170 g for

10 min at 4°C. The nuclei were resupended in 5 ml of RIPA/gram of embryos and split into 1-ml aliquots. Chromatin was sonicated using a Covaris S220 (100 W Peak Power, 20% Duty Factor, 200 Cycles/Burst, 15 min total time) and insoluble material was removed by centrifugation at 13.2 krpm for 20 min (4°C). Soluble chromatin was pre-cleared by adding RIPA-equilibrated 50% slurry of protein A+G (1:1) sepharose beads and rotating at 4°C for 1 h. Then 200 μl of chromatin was incubated overnight with 4 μl of the respective antibody: αACF1 Rb2 (22), αFLAGm2 (Sigma, Cat. No. F1804), and αISWI Rb1 (Becker Lab, unpublished). Then 30 μl protein A+G (1:1) 50% slurry was added and the tubes were rotated for 3 h at 4°C. After five washes with RIPA buffer, RNase-A was added (10 μg/100 μl; Sigma, Cat. No. R4875) and the tubes were incubated at 37°C for 20 min. Subsequent protease digestion (using 250 ng/μl Proteinase K; Genaxxon, Cat. No. M3036.0100) and cross-link reversal were performed simultaneously at 68°C for 2 h. DNA was purified using 1.8× Agencourt AMPure XP beads (Beckman Coulter, Cat. No. A63880) following a standard protocol and eluted in 50 μl of 5 mM Tris–Cl, pH 8. Purified DNA was used for standard qPCR analysis at 1:2 dilution. The primers are listed in Table S3.

### RT–qPCR

For *LacZ* expression analysis, embryos were collected 2–8 h AEL and dechorionated in 25% bleach for 3 min. After extensive washes with PBS, the embryos were transferred into a 1.5-ml tube, resuspended in 300 μl of QIAzol (QIAgen, Cat. No. 79306), and homogenized using a metal pestle. After addition of 700 μl of QIAzol, the samples were snap-frozen in liquid nitrogen and stored at –80°C until further processing. RNA was extracted using the standard protocol provided by QIAgen. The Superscript III First Strand Synthesis System (Invitrogen, Cat. No. 18080051, random hexamer priming) was used to generate cDNA starting from 1.5 μg of total RNA. cDNA was used for standard qPCR analysis at 1:10 dilution. The primers are listed in Table S3.

### Immunofluorescence microscopy

For immunofluorescence of *Drosophila* Kc167 cells, 200 μl of cells (>$10^6$ cell/ml) were transferred onto polylysine–coated three-well depression slides (Thermo Scientific, Cat. No. 631-0453) and incubated for 1.5 h at 26°C. The cells were washed with PBS and fixed in PBS/3.7% formaldehyde for 10 min. After two washes with PBS, the cells were permeabilized in ice-cold PBS/0.25% Triton X-100 for 6 min. The cells were washed twice with PBS and blocked with PBS/0.1% Triton X-100/5% normal donkey serum (Jackson Immuno Research)/5% nonfat milk for 2 h. After a brief wash with PBS, the cells were incubated overnight at RT with primary antibodies αV5 (1:1,000, GenScript, Cat. No. A00623) and αmCherry (43) (1:20). The cells were washed twice with PBS/0.1% Triton X-100 and incubated with secondary antibodies donkey-αrat-Cy3 (1:500, Jackson Immuno Research) and donkey-αrabbit-Alexa488 (1:300, Jackson Immuno Research) for 2 h at RT. The cells were washed twice with PBS/0.1% Triton X-100, incubated with 1:500 DAPI for 10 min at RT, and washed again with PBS. Coverslips were mounted using Vectashield mounting medium (Vector Laboratories, Cat. No. H-1000) and sealed with nail

polish. Pictures were acquired on a Leica Sp5 confocal microscope using the same settings for all the constructs.

### Artificial tethering of ACF1 to multiple reporters in *Drosophila* cells

Kc167 cells containing the barcoded reporter library were generated as previously described (24). Plasmids for the expression of GAL4-ACF1 fusion and controls were derived from pAc5-Gal4-V5-HP1a-T2A-mCherry (24) by Gibson assembly. All constructs were validated using DNA sequencing and restriction digestion analysis. The artificial tethering, including sample preparation, sequencing, and data processing/analysis, was performed as described in reference 24, including GAL4-HP1 construct as a positive control. Two biological replicates for each condition were analyzed. Reporters with normalized counts equal to zero in at least one condition were discarded. Linear models were calculated using the *lm* function in R. For transient expression of Gal4-ACF1 fusion and controls, 3 × $10^6$ Kc167 cells were transfected with 1 μg of the corresponding plasmid using the X-tremeGENE HP transfection reagent (Sigma, Cat. No. 6366236001) following the standard protocol (4.5:1 transfection reagent:DNA ratio). 3 d after transfection, 1 ml cells was collected, spun at 800 g for 5 min, resuspended in 20 μl of 5× Laemmli sample buffer per $10^6$ cells, and boiled at 95°C for 10 min.

### Single-embryo RNA-seq

Before RNA-seq, the *Acf*$^C$ flies were backcrossed with the wild-type *OrR* strain for eight generations. Embryos were collected 0–45 min AEL and allowed to develop at 25°C until approximately 30 min before the desired stage (around 1 h for Bownes Stage 3 and 4 h for Bownes Stage 8). Without prior dechorionation, the embryos were hand-picked and submerged into a drop of Voltalef 10 S halocarbon oil (Lehman and Voss Co.) placed on a microscope slide. After about 5 min, the embryonic structures became visible under the stereomicroscope. Embryos were allowed to develop further under the halocarbon oil until the desired stage. Single embryos were picked and crushed with a 26-G needle into 200 μl of lysis buffer (supplemented with Proteinase K) from the Agencourt RNAdvance Tissue kit (Beckman Coulter, Cat. No. A32645). After the addition of 10 μl of 1:100 ERCC Spike-in RNA mix (Ambion, Cat. No. 4456740), the samples were incubated at 37°C for 20 min, snap-frozen in liquid nitrogen, and stored at –80°C. Total RNA was extracted from the single-embryo homogenate using the same Agencourt RNAdvance Tissue kit, following standard protocol but using half of the volumes recommended. RNA integrity was checked on a Bioanalyzer 2100 instrument (Agilent). Ribosomal RNA depletion was achieved using an rRNA depletion kit (human/mouse/rat) (New Englands Biolab, Cat. No. E6310) and the rRNA-depleted RNA was stored at –80°C until further processing. Nondirectional libraries were prepared using an NEBnext Ultra RNA Library Prep kit for Illumina (New Englands Biolab, Cat. No. E7530S) following standard protocol. Six replicates per genotype and stage were sequenced on an Illumina HiSeq1500 instrument. Paired-end RNA-seq reads were mapped against the reference genome (FB2016_01 dmel_r6.09 with selected chromosomes) using STAR (version 2.5.0a) with *quantMode Gene-Counts* for counting reads per gene (44). One replicate from the *Acf*$^C$

genotype (Stage 3) was excluded because of improper staging (data not depicted). Size factors for normalization were calculated by DESeq2 (45). PCA was carried out on selected genes with variance across samples between the 85th and 99th percentiles. Genes with a read count equal to zero in at least half of the samples were filtered out for further analysis. Differential expression (DESeq2) analysis (mutant versus wild type) was carried out by fitting negative binomial GLM independently for the two developmental stages (45). Cutoffs for adjusted P-values were defined at the 0.1 level. Full lists of differentially expressed genes are presented in Tables S1 and S2. Gene ontology analysis on significantly different genes was performed on the FlyMine online database (46). Genes were assigned to five-state chromatin domains (25) by the *nearest* method from the GenomicRanges Bioconductor packages. Trends on MA plot were visualized by local polynomial regression fitting (*loess*). modENCODE histone modification signals (smoothed M-values) (26) were averaged over genes and low/high levels were distinguished by a cutoff based on the local minimum in the density of the H3K36me3 levels. Genes were classified as marked/unmarked based on whether they carry high/low histone modification levels in all four marks investigated in the analysis.

### Nucleosome mapping and autocorrelation

For mapping nucleosomes, embryos were collected 2–8 h AEL. The embryos (between 0.2 and 0.5 g per replicate and genotype) were dechorionated and fixed as described in the ChIP-qPCR section.

For nuclei isolation, the embryos were slowly thawed and dounced using a glass homogenizer (Schubert, Cat. No. 9164693) with 20 strokes each of the A and B pestles in ice-cold NX-I buffer (15 mM Hepes, pH 7.6, 10 mM KCl, 2 mM MgCl$_2$, 0.5 mM EGTA, 0.1 mM EDTA, 350 mM sucrose, 1 mM DTT, 0.2 mM PMSF, and Roche cOmplete Protease inhibitor without EDTA). Nuclei were subsequently pelleted at 3,500 rpm for 10 min at 4°C. For MNase digestion, the nuclei were suspended in the RIPA buffer supplemented with 2 mM CaCl$_2$. The nuclei were digested with 13 units of MNase per gram of starting embryos (Sigma, Cat. No. N5386) for 15 min at 37°C while shaking at 500 rpm. The reaction was stopped by adding 0.5 M EDTA (pH 8.0) to a final concentration of 10 mM and the tubes were quickly transferred to ice for 5 min. The nuclei were spun at 12.5 krpm for 10 min at 4°C. The supernatants containing most of the DNA were collected and the residual RNA was digested by RNase-A (50 μg/ml; Sigma, Cat. No. R4875) at 37°C for 30 min. Protein digestion and cross-linking reversal were performed as previously described in the ChIP-qPCR section. DNA was purified using 1.8× Agencourt AMPure XP beads (Beckman Coulter, Cat. No. A63880) following standard protocol and eluted in 50 μl of 5 mM Tris–Cl, pH 8. Recovered DNA was quantified using the Qubit dsDNA HS assay kit (Life Technologies, Cat. No. Q32851) and sequencing libraries were prepared using a custom-made protocol available upon request. Libraries were sequenced on a HiSeq 1500 (Illumina) instrument.

Paired-end reads were mapped to *Drosophila* genome version dm6. We used Bowtie v1.1.1 with "-X 750" parameter setting. Dyad coverage vectors were obtained by size-selecting fragments of length >120 and <200 bp and resizing their length to 50 bp fixed at the fragment center.

The nucleosome dyad maps used for autocorrelation were generated, validated, and interpreted by Jain et al (27 *Preprint*). The autocorrelation function was calculated for the dyad coverage vectors obtained for the entire genome and for the five-state domains described by Filion et al (25). The vectors for the last cases represent head-to-tail concatemerized regions of given annotation. The function was run for the lag length of 1,000 bp. Nucleosomal repeat lengths were obtained by linear regression of the first and second autocorrelation peak positions with zero intercept. The slope of the regression was defined as repeat length. Values reported in the text correspond to average nucleosomal repeat length (between biological replicates) ± SEM.

### Accession codes

Sequencing data have been deposited in the Gene Expression Omnibus under accession numbers GSE106759 (artificial tethering of ACF1 in Kc167 cells) and GSE106733 (nucleosome maps and single-embryo RNA-seq).

## Supplementary Information

## Acknowledgements

We thank K Förstemann for sharing reagents and cell lines for gRNA testing. We thank S Krebs and H Blum and the Netherlands Cancer Institute Genomics Core Facility for outstanding sequencing service. PB Becker and A Scacchetti are funded by the European Research Council (ERC), MSCA-ITN-2014-ETN No. 642934. F Schnorrer is funded by the ERC [(FP/2007–2013); grant 310939], the Centre Nationnal de la Recherche Scientifique, the excellence initiative Aix-Marseille University AMIDEX, the Agence Nationale de la Recherche, and the LabEX-INFORM. B van Steensel is supported by National Institutes of Health grant U54 DK107965.

### Author Contributions

A Scacchetti: Conceptualization, formal analysis, investigation, methodology, writing—original draft, review, and editing.
L Brueckner: Formal analysis, investigation, methodology, writing—review and editing.
D Jain: Formal analysis, investigation, methodology, writing—review and editing.
T Schauer: Data curation, formal analysis, methodology, writing—review and editing.
X Zhang: Methodology, writing—review and editing.
F Schnorrer: Supervision, funding acquisition, writing—review and editing.
B van Steensel: Supervision, funding acquisition, writing—review and editing.
T Straub: Data curation, formal analysis, methodology, writing—review and editing.
PB Becker: Conceptualization, supervision, funding acquisition, writing—original draft, project administration, writing—review and editing.

## Conflict of Interest Statement

The authors declare that they have no conflict of interest.

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
