## [Reviewer comments · Life Science Alliance]

CHRAC/ACF Contribute to the Repressive Ground State of Chromatin

Alessandro Scacchetti, Laura Brueckner, Dhawal Jain, Tamas Schauer, Xu Zhang, Frank Schnorrer, Bas van Steensel, Tobias Straub and Peter B. Becker

DOI: 10.26508/lsa.201700024

Review timeline:	Submission date:	24 January 2018
	Revision received:	24 January 2018
	Editorial Decision:	25 January 2018

Report:

(Note: Letters and reports are not edited. The original formatting of letters and referee reports may not be reflected in this compilation.)

1st Revision – authors' response

24 January 2018

REFeree REPORTS

Referee #1:

The ACF and CHRAC chromatin remodeling enzymes represent a pair of the original enzymes identified in early biochemical studies. Both enzymes can mobilize nucleosomes in cis, and ACF was identified and characterized for its ability to promote nucleosome assembly and spacing in the presence of the NAP1 histone chaperone. Each enzyme contains the ISWI catalytic subunit and the ACF1 auxiliary subunit. CHRAC also contains two additional histone fold-containing subunits, CHRAC14 and 17. There have been several studies describing the phenotype of *Drosophila* that lack a functional ACF1 gene. Early work from the Kadonaga group (Fyodorov et al., 2004) used two alleles of ACF1 - Acf1(1) and Acf1(2). Whereas the Acf1(1) allele has subsequently been shown to express a C-terminal truncation product, the Acf1(2) allele would appear to be a true null (lacks promoter, ATG, and first exon). These genetic studies demonstrated that loss of ACF1 led to a semi-lethal phenotype and delayed development. Survivor flies, however, were fertile and appeared phenotypically normal. Defects in heterochromatin structure and bulk nucleosome spacing were observed. To date, there have been no analyses of genome-wide transcriptional defects that this reviewer is aware of.

Response: Indeed, such an analysis had not been done.

In this manuscript, the authors perform two types of analyses to further characterize the functioning of ACF1 (and consequently ACF and CHRAC): (1) they use Gal4 fusions to tether Acf1 to a large number of reporter loci; and (2) they perform RNA-seq and nucleosome mapping experiments on an Acf1 null strain (a CRISPR/Cas9-mediated deletion and a null created by imprecise transposon excision as per previous studies). The authors report that tethering Acf1 adjacent to a reporter gene leads to a ~2-fold repression of poorly transcribed reporters and repression correlated with the chromatin state of the reporter loci; and (2) loss of Acf1 in embryos led to a depression of a small subset of poorly expressed genes, and these affected genes were again characterized by chromatin states associated with developmental- and polycomb-regulated states. Loss of Acf1 also changed the

spacing of nucleosomes (larger repeat length), consistent with previous studies (and work in yeast for *Iswi* mutants). Consistent with previous studies demonstrating that *Acf1* mutant flies show little phenotype (at least those that survive), the current results show little impact on the transcriptome. Most likely this is due to redundancy with other ISWI-like remodeling enzymes, as suggested by the authors. The observation that *Acf1* contributes to nucleosome spacing is also consistent with previous work in *Drosophila* and also yeast. Demonstrating that *Acf1* contributes to transcriptional repression is a valuable contribution. There are several issues that detract from this study:

1. In the Gal4-*Acf1* tethering studies, it is not clear if repression of reporters is due to nucleosome remodeling activity or just due to some type of blocking behavior. Notably, the authors do not observe changes in nucleosome positions at target reporters. I believe that the protein-protein interface for interaction between *Acf1* and ISWI is known? Tethering of such an allele would make these studies much stronger. Alternatively, perhaps the tethering experiment can be performed in cells that lack or are depleted for ISWI?

Response: We thank the referee for suggesting further experiments aimed to clarify the mechanism of repression by GAL4DBD-ACF1. Based on the current data we cannot formally exclude that repressive effects are not due to nucleosome ‘remodeling’. The fact that the catalytic subunit, ISWI, is recruited to the tethering site suggests that a complete nucleosome remodeler is enriched. Depleting ISWI will not clarify the issue since ISWI is part of several remodeling complexes. Tethering mutated ACF1 will not overcome the principal concern that a tethered remodeler with a novel DNA binding domain is an artificial situation to start with. We alert the reader to the ‘artificial nature of the tethering experiment’ on Page 6, and have now expanded the sentence to increase the readers’ awareness. The tethering experiment with all its weaknesses prompted a careful analysis of our transcriptome data and a focus on lowly transcribed regions, which would have been overlooked in any standard analysis. The fact that in the loss-of-function study in embryos the transcriptional de-repression correlates with broad disruption of the nucleosome fiber supports the idea that indeed the nucleosome remodeling activity of ACF is involved.

2. The authors explain in detail that the previous genetic studies on *Acf1* relied heavily on the *Acf1(1)* allele that encodes a truncation. However, it appears that identical data was obtained by the Kadonaga group using the *Acf1(2)* allele which appears to be a null. This needs better explanation. The author also need to do a more comprehensive job at comparing their mutant phenotype to these previous studies. Previous work (from the Becker lab) did quite an extensive analysis).

Response: We had tested the *Acf1(2)* allele in Börner et al. [1]. The results revealed an oogenesis phenotype like the *Acf1(1)* allele, which is not ‘null’. Both alleles are very similar in that they represent relatively small deletions in the 5’ end of the ACF1 gene. We extensively compared the *Acf1(7)* allele to *Acf1(1)* and *Acf1(2)* in this earlier work and concluded that it corresponds to a complete loss of function. We now mention *Acf1(2)* along with *Acf1(1)* in the introduction (page 3) and added a comment to the method section (page 12, 14) explaining this issue better. We independently had initiated the CRISPR deletion of the entire gene (*Acf^c*) and want to use the opportunity to present this novel tool to the fly community. To the best of our knowledge, both *Acf1(7)* and *Acf^c* are loss of function alleles that we can use interchangeably.

3. Figure 3. Here it would be helpful to show an expression plot of WT compared to mutant, so that differences can be directly compared. Such a plot would also potentially emphasize that only poorly expressed genes are impacted. Additionally, a volcano plot could be shown to emphasize significant changes in gene expression compared to WT. Currently, there are little panels that actually show the number of genes that change.

Response: In Figure 3 c,d we show the log₂ fold-change (mutant/wild-type) in relation to the mean log₂ expression of every gene, highlighting statistically significant genes (red dots). We think this is very similar to what the referee is asking. S/he may have overlooked these data.

4. On page 4, top paragraph, the authors state "some phenotypes observed in *Acf1*(1) could not be reproduced in a larger gene deletion..." Is this statement citing a published observation or unpublished? It is not clear as written.

Response: We refer to the published work of Börner et al. [1] and have now added some text on page 3 for better explanation.

5. Page 10, middle. "...giving rise to unclear gain-of-function effects..). I don't see the evidence for this statement in the Boerner paper? Is this referring to the potential for gain-of-function or was this actually demonstrated?

Response: This is indeed referring to a potential gain-of-function as discussed in [1]. We modified our statement in the discussion on page 10 and encourage the interested reader to consult the paper where the phenotypes of the alleles were compared.

Referee #2:

This manuscript describes the role of the CHAC/ACF chromatin remodeler in gene repression in *Drosophila*. Authors provide three major forms of evidence supporting their claims:

- 1) Analysis of reporter gene expression following artificial recruitment of ACF1 to multiple reporters integrated throughout the fly genome.
- 2) RNA-Seq data for *acf1* mutant embryos.
- 3) Reanalysis of previously-published nucleosome mapping data.

This manuscript provides some novel information for readers interested in chromatin control of transcription, and in my opinion will be a valuable contribution and should be published.

Response: We thank the referee for the appreciation.

I have a few technical concerns which I hope could be addressed in a revised manuscript. The most interesting and novel aspect of this manuscript is the use of hundreds of reporter genes integrated throughout the fly genome to assay the effects of artificial ACF1 recruitment. The authors document a decrease in expression of reporters located in BLACK and BLUE (repressive) chromatin environments in the presence of appropriately targeted Gal4-Acf1. However, the untargeted Acf1 control also revealed some repressive activity in these reporters. My concern is that the enhanced activity of the Gal4-Acf1 fusion might not result from Gal4 DBD targeting, but rather from higher expression of this fusion relative to the Acf1 alone control.

Response: We thank the referee for pointing out this interesting aspect. Nucleosome remodelers can work if tethered [2], but given the dynamic nature of DNA interactions observed with most transcription factors we think that the tethering rather increases the local concentration of the factor around the UAS^{Gal} site. Accordingly, a global increase of the ACF1 concentration will have the same effect, but milder. Importantly, the selective effect of ACF1 on chromatin environment and expression strength is not altered, whether the global or local concentrations of ACF1 are increased. All factors, including HP1, are expressed from the same strong Actin5C promoter. In their original paper, Brueckner et al. [3] also observed that overexpressing untethered HP1 had a mild repressive effect, but the effect of expressing tethered HP1 was more pronounced. The situation for ACF1 is very similar. We now expanded the discussion reflecting on these issues (page 11).

In addition to more careful quantitation of the levels of Acf1 and Gal4-Acf1 (Figure S2B is not quantitative), the authors could further alleviate this concern by examining a handful of individual reporters in cell lines overexpressing the two Acf1 constructs - ideally, overexpression of the untargeted Acf1 would not repress these reporters as well as moderate expression of the Gal4-

targeted version. I think this is an important experiment to support the central conclusion of the study.

Response: Considering that we transiently transfect a pool of many clones we cannot be certain about the expression levels in individual clones. This weakness of the experimental set-up translates into an advantage since working with such a population averages out the individual, anecdotal effects. Our conclusions are based on statistical arguments. By examining individual cell lines (how many? how selected?) we would end up presenting anecdotes of unclear representation of the overall picture. For our conclusion, it is not important how (local, global) and to which extent the concentration of ACF1 (or HP1) is increased. Importantly, the effect is selective for black/blue chromatin, and is later confirmed by the complementary mutational approach.

With regards to the transcriptome analysis of early embryos, the authors have focused entirely on annotated coding regions. Given that disorganized chromatin is sometimes associated with increased levels of noncoding transcripts, or altered transcript structure, it would be nice to see a more detailed analysis of transcription across normally silent regions of the genome.

Response: This is true. We looked at the de-repression of transposons, which is usually a sensitive measure of chromatin perturbation and did not find any, presumably because they are repressed by proper heterochromatin. This is now mentioned in the discussion (page 12). The occurrence of cryptic or antisense expression usually requires to do the analysis in exosome mutants, since these spurious transcripts are rather unstable. Such mutants are not available in flies. Because our transcriptome analysis was performed on single *Drosophila* embryos the relatively low number of reads does not permit a deeper investigation of very rare and unannotated transcriptions.

Finally, the analysis of nucleosome positioning is entirely based on autocorrelation functions. As a further visualization, I would appreciate seeing a few metagenes for genes in the various chromatin colors showing the raw nucleosome occupancy data, aligned perhaps by +1 nucleosomes. This should also show the altered nucleosome repeat length in a more intuitive way than the autocorrelation functions.

Response: We considered presenting these TSS-centered metagene plots, but they would remain uninformative. Nucleosomes are well-phased at active promoters, but this alignment is not influenced by ACF1 deletion (unpublished observation), in agreement with the observation that ACF1 does not affect red/yellow promoters. The poorly expressed genes that turn out to be ACF1 targets do not show visibly appreciative nucleosome phasing at the TSS. We rather quantified the changes in nucleosome phasing across different chromatin states using a more relevant and sensitive approach: the autocorrelation function. The autocorrelation function measures periodicity along the entire length of the genomic region and does not suffer from centering of the data or averaging of signal values as we typically do while plotting TSS-centered metagene plots. This measure best represents subtle changes in our data describing ACF1-dependent nucleosome effects.

Referee #3:

In this manuscript, Scacchetti et al. investigate the functional role of ACF1 containing nucleosome remodeling complexes in genome regulation. While such complexes have been intensively studied, the lack of true null mutations and the inability of performing ChIP experiments have limited the insights into the biological role of these nucleosome remodelers. The authors used two different approaches to tackle this problem. They tethered ACF1 to many different sites of the genome and deduced the transcriptional response of these regions. Their results indicate that the effect of ACF1 is depending on the chromatin context and especially prominent at lowly expressed genes in overall inactive chromatin domains. Secondly, the authors determined the transcription and nucleosome distribution profiles of embryos where ACF1 has been deleted. Here, it was found that loss of ACF1 leads to a widespread de-repression of low transcribed regions.

Overall, the authors conclude that ACF1 containing remodeling complexes contribute to a "physiological chromatin regularity and general repression". While this main conclusion of the work

is well supported by the data and by experiments that contain the necessary controls, it is not entirely clear which features distinguish areas of the genome that are responsive to ACF1-containing nucleosome remodeling complex from regions that are independent of these factors. In light of no clear targeting mechanisms of ACF1 containing nucleosome controlling complexes, this becomes an important question that needs to be addressed. The authors suggest that specific gene regulation via for example histone modifications and specific remodelers are superimposed to this ground state. While this is an attractive idea, there is no evidence on such hierarchy of events.

Criticism

The title of the paper suggests a 'ground state of chromatin'. In the discussion, the authors state: 'The impact of ACF1 depletion was more evident for inactive chromatin domains, establishing a clear correlation between the extent of physiological chromatin regularity and general repression, which we suggest is of causal nature.' Based on the title, I would expect such ground state to be causally linked to the function of the ACF1 containing remodeling complexes.

Response: We think the referee may have misunderstood our point. Our title is not about a 'ground state of chromatin' (which is the nucleosome fiber), but about a 'ground state of repression' (acknowledging the fact that 'higher order repression' involve specific factors such as HP1, polycomb and alike). Our point is that this 'repressive ground state' of chromatin is indeed implemented by the regularity of the nucleosome fiber. It is only seen when more elaborate repression mechanisms are not at work. This line of thought is detailed in the concluding paragraph of the discussion.

- The rationale for switching between the ACF⁷ and ACF^C null mutants in the different types of analyses is not clear.

Response: ACF^f and ACF^f are both true null mutants and their embryonic phenotypes (decreased larval hatching) are very similar ([1]; this study). We characterized the ACF^f allele and independently initiated the generation of a CRISPR allele. To the best of our knowledge, both ACF^f and ACF^f are loss of function alleles that we can use interchangeably. We want to use the opportunity to present the novel tool/allele to the fly community. We modified the text at various places to explain these facts (see above).

- To deduce a role for ACF1 containing nucleosome remodeling complexes in setting up the 'ground state of chromatin' it is imperative to analyze the chromatin environment of selected loci after tethering ACF1-GAL4DBD. Do these show the same changes that the authors deduce for the ACF1 dependent loci in the genome wide analysis of ACF1 mutant embryos? In absence of such data, other functions of ACF1 outside of nucleosome remodeling complexes need to be considered for explaining the observations made.

Response: As mentioned above, we do not 'deduce that nucleosome remodeling complexes set up a ground state of chromatin', but rather conclude that they maintain a 'ground state of repression'. This conclusion follows from the statistical analysis of large numbers of genes, grouped according to chromatin types. The population of reporter genes contains the same reporter cassette in different chromatin environments. We would have to work with clonal lines of individual insertions, which would yield a lot of anecdotal data, of unclear representation. Therefore, we do not think that such an analysis would be informative.

- From the analysis and discussion, it is not entirely clear what molecular parameters separates genome regions that are susceptible to loss of ACF1 or that respond to ACF1 tethering with changes in transcription.

Response: In the Drosophila field the classification of chromatin states by Fillon et al. [4] is very well established and used in numerous publications. Each genomic location has been classified according to the five principal chromatin states and we have used this classification, as is apparent from our description of the methods.

- In light of the suggestion that ACF1 containing nucleosome complexes contribute to general repression, the context dependence observed for ACF1 effects in the tethering experiments as well as the small number of genes significantly affected by loss of ACF1 seem surprising. Overall, it appears that there must be other components besides ACF1 containing nucleosome remodeling complexes contributing to the observed effects.

Response: As mentioned above, we do not argue that ACF1 is involved in general repression, but in a ground state of repression. Of course, many specific and dedicated repressors are known that work in prominent repression pathways (Heterochromatin, Polycomb, etc.). We argue that there is a level of repression (we call it the ground state) that occurs when specific repressors are not involved. We indeed think that this is installed by a regular nucleosome fiber. The reviewer is right in that the effects are small. This may, at least in part, be due to functional redundancy with other nucleosome spacing complexes, such as RSF. We mention this fact in the discussion.

Specific comments

- General figure layout: The arrangement of panels in figures is very crowded making it hard to look at the different results presented. Some of the labeling of panels is done in varying fonts sizes. Some of the font sizes in figure are very small, likely making it difficult to read the labeling at final reproduction size. Lastly, similar data that are presented in the same type of graph or blot are sized differently in the different panels within the main and supplemental figures.

Response: We have tried to improve the presentation and hope that it is acceptable now.

- It would help to identify the different bands in Western blots besides pointing out their identity in the figure legend.

Response: The correct bands are now labeled with asterisks.

- The ChIP-qPCR results (Fig. 1c) are presented with error bars reflecting SEM. This seems highly unusual. Indeed, STDEV represents the variability of the results obtained in the biological replicates.

Response: The SEM is the inferential error, which indicates how close the sample mean is to the population mean (the 'true' value). This is the information to be communicated if experiments are considered representative of an infinite group of similar measurements. These errors provide information about the robustness the observed biological phenomena. The standard deviation is frequently used in such depictions, however, this value does not provide information about the population. It simply reflects the scatter of the measurements in the sample. There is no information about the population as provided in SEM. Inferential error (SEM or CI) bars are preferred and even explicitly recommended by various journals. We will not change for an inferior metric in our displays.

- The text will benefit from careful proofreading. There are several typos and spelling errors.

Response: We apologize for the oversight and have carefully re-edited the text.

References cited in this rebuttal:

1. Boerner K, Jain D, Vazquez-Pianzola P, Vengadasalam S, Steffen N, Fyodorov DV, Tomancak P, Konev A, Suter B, Becker PB (2016) A role for tuned levels of nucleosome remodeler subunit ACF1 during *Drosophila* oogenesis. *Developmental biology* **411**: 217-230
2. McKnight JN, Jenkins KR, Nodelman IM, Escobar T, Bowman GD (2011) Extranucleosomal DNA binding directs nucleosome sliding by Chd1. *Molecular and cellular biology* **31**: 4746-4759
3. Brueckner L, van Arensbergen J, Akhtar W, Pagie L, van Steensel B (2016) High-throughput assessment of context-dependent effects of chromatin proteins. *Epigenetics Chromatin* **9**: 43

4. Filion GJ, van Bommel JG, Braunschweig U, Talhout W, Kind J, Ward LD, Brugman W, de Castro IJ, Kerkhoven RM, Bussemaker HJ, *et al.* (2010) Systematic protein location mapping reveals five principal chromatin types in *Drosophila* cells. *Cell* **143**: 212-224

1st Editorial Decision

25 January 2018

Thank you for submitting your revised manuscript entitled "CHRAC/ACF Contribute to the Repressive Ground State of Chromatin" to Life Science Alliance.

The manuscript was previously reviewed at a different journal and the referee reports have been transferred to Life Science Alliance. You provided a revised manuscript and a detailed point-by-point response to the reports obtained during peer-review elsewhere, and both the academic editor and I appreciate the response and the introduced changes. We are thus happy to accept your manuscript in principle for publication in Life Science Alliance.

Thank you for this very nice contribution to Life Science Alliance and for your enthusiasm for our new open-access journal! I am very much looking forward to publishing your paper.